# L-Arabinose Alters the *E. coli* Transcriptome to Favor Biofilm Growth and Enhances Survival During Fluoroquinolone Stress

**DOI:** 10.3390/microorganisms13071665

**Published:** 2025-07-15

**Authors:** Katherine M. Austin, Jenna K. Frizzell, Audrey A. Neighmond, Isabella J. Moppel, Lisa M. Ryno

**Affiliations:** Department of Chemistry and Biochemistry, Oberlin College, Oberlin, OH 44074, USA

**Keywords:** biofilm, *E. coli*, fluoroquinolone, transcriptomics

## Abstract

Environmental conditions, including nutrient composition and temperature, influence biofilm formation and antibiotic resistance in *Escherichia coli*. Understanding how specific metabolites modulate these processes is critical for improving antimicrobial strategies. Here, we investigated the growth and composition of *Escherichia coli* in both planktonic and biofilm states in the presence of L-arabinose, with and without exposure to the fluoroquinolone antibiotic levofloxacin, at two temperatures: 28 and 37 °C. At both temperatures, L-arabinose increased the growth rate of planktonic *E. coli* but resulted in reduced total growth; concurrently, it enhanced biofilm growth at 37 °C. L-arabinose reduced the efficacy of levofloxacin and promoted growth in sub-minimum inhibitory concentrations (25 ng/mL). Transcriptomic analyses provided insight into the molecular basis of arabinose-mediated reduced susceptibility of *E. coli* to levofloxacin. We found that L-arabinose had a temperature- and state-dependent impact on the transcriptome. Using gene ontology overrepresentation analyses, we found that L-arabinose modulated the expression of many critical antibiotic resistance genes, including efflux pumps (*ydeA*, *mdtH*, *mdtM*), transporters (*proVWX*), and biofilm-related genes for external structures like pili (*fimA*) and curli (*csgA*, *csgB*). This study demonstrates a previously uncharacterized role for L-arabinose in modulating antibiotic resistance and biofilm-associated gene expression in *E. coli* and provides a foundation for additional exploration of sugar-mediated antibiotic sensitivity in bacterial biofilms.

## 1. Introduction

*Escherichia coli* is a significant human pathogen responsible for approximately 265,000 illnesses and about 100 deaths in the United States each year [1]. *E. coli* causes pneumonia, urinary tract, and bloodstream infections, particularly in hospitalized patients. Pathogenic strains of *E. coli* are transmitted through contaminated food sources, leading to foodborne diseases through colonization of the gastrointestinal tract. A fundamental aspect of *E. coli* biology is its ability to strategically fluctuate between the planktonic (free-swimming) and biofilm (surface-attached) states to gain ecological advantages [2]. The planktonic state is composed of individual motile cells. In contrast, biofilm is a sedentary community of bacteria embedded in an extracellular polymeric substance (EPS), a protective matrix of self-produced macromolecules such as proteins, carbohydrates, and other polymers [3]. The EPS promotes survival by providing a physical barrier that protects bacterial cells from antibiotics and other environmental stressors.

Biofilm development is a well-characterized multi-step dynamic process. Biofilm growth initiates with the reversible, non-specific attachment of planktonic cells to a surface, followed by irreversible binding as interactions occur between the bacterial colonies and the surface. Extracellular adhesins, such as curli and fimbriae, facilitate this binding, promoting stronger bacterium–substrate interactions and robust biofilm formation [2]. As the biofilm matures, the production of the EPS becomes prominent and contributes to specific interactions, including hydrogen bonding and ionic and hydrophobic interactions, further establishing the biofilm state. Interconnected environmental factors such as available nutrients, pH, temperature, and salt concentrations play a critical role in modulating biofilm formation and composition [3,4]. Finally, biofilms may enter a dispersal phase, where bacterial cells can detach from the matrix and navigate the surrounding environment to seek new surfaces for colonization, allowing the biofilm growth cycle to repeat [2].

Among factors that can influence *E. coli* growth and physiology, available carbon sources play a central role. Carbon sources are imported and broken down to supply the bacterial cell with amino acids and adenosine triphosphate (ATP) for various biological processes. Bacteria hierarchically prioritize the metabolism of carbohydrates based on their availability in the environment; the cell preferentially metabolizes the sugar that yields the highest energy output. Environmental sugars such as glucose, arabinose, xylose, and lactose—which *E. coli* use as energy sources—play significant roles in ecological and biological processes. Glucose is the most efficiently metabolized sugar and is thus prioritized, often inhibiting the uptake of other carbohydrates. Lactose, a disaccharide composed of D-glucose and D-galactose, is next in the hierarchy due to fewer metabolic steps and higher energy yield compared to arabinose and xylose, resulting in the repression of genes involved in the use of these sugars when lactose is available [5,6]. Similarly, arabinose inhibits xylose metabolism by competitively binding to the promoter of xylose genes, blocking transcription and leading to catabolite repression [7]. It is through similar interactions and additional mechanisms that the environment of the microorganism, including the availability of nutrient sources like sugars, significantly impact the transcriptome [8,9,10,11]. Modifications in the organism′s transcriptome affect metabolic and signaling pathways, resulting in phenotypic variations [9,12]. In some cases, these phenotypic variations can lead to increased fitness and survival, while others can increase the organism’s sensitivity to external insults.

We focus this study on the sugar L-arabinose. This monosaccharide is a primary component of hemicellulose, a key polysaccharide in plant cell walls and found abundantly in corn, beets, and wheat bran [13]. *E. coli* maintains two independent transport mechanisms active at a basal level to intake arabinose into the cell. The primary mechanism is a proton symport permease, AraE. This method uses an electrochemical proton gradient to drive the active transport of L-arabinose into the cell. An additional mechanism of arabinose import is via the protein complex AraFGH, an ATP-binding transporter. AraF is a periplasmic binding protein with a high affinity for L-arabinose, AraH is a trans-membrane protein, and AraG is a soluble protein containing the ATP binding site for active import [14,15]. While AraFGH exhibits a strong affinity for transport, AraE is the favored mechanism for conserving cellular energy when L-arabinose is available in high concentrations.

Our lab previously determined that the efficacy of the fluoroquinolone antibiotic levofloxacin was impacted by the presence of L-arabinose, increasing the required minimum inhibitory concentration of the antibiotic 3-fold and the tolerance of the bacteria 4-fold [16]. Here, we examined the independent influence of L-arabinose at different growth temperatures on the biofilm-forming *E. coli* strain PHL628, which is derived from the parent strain K-12 MG1655 and has a mutation in the *ompR* gene that leads to constitutive activation and biofilm formation [17]. We explored planktonic and biofilm growth and composition with L-arabinose and in the presence or absence of the fluoroquinolone levofloxacin at a growth temperature that is optimal for biofilm formation (28 °C) and human physiological temperature (37 °C). Additionally, we analyzed changes to the transcriptome with arabinose at these temperatures to understand what gene expression changes occurred in both the planktonic and biofilm states that might shed light on the decreased sensitivity of *E. coli* to levofloxacin in the presence of arabinose.

## 2. Materials and Methods

### 2.1. Bacterial Strain Selection and Cultivation

Kanamycin (Sigma-Aldrich, St. Louis, MO, USA) was used to select for the PHL628 *E. coli* cells (a kind gift from Anthony G. Hay, Ph.D., Cornell University) for all experiments [17]. Then, 10% (*w*/*w*) L-arabinose (Sigma-Aldrich) stock solutions were made by dissolving 1.00 g of L-(+)-arabinose in 9.00 mL of ultrapure water and sterile filtering the solution using a 0.2 µm polyethersulfone membrane syringe filter (VWR, Radnor, PA, USA). In total, 50 mg/mL (1000×) kanamycin was prepared in ultrapure water and sterile filtered prior to use. Then, 2 mg/mL levofloxacin (Sigma-Aldrich) stock solution in ultrapure water was prepared and then diluted to 20 µg/mL and sterile filtered.

All *E. coli* cell cultures were grown in lysogeny broth (LB), composed of 25.00 g of LB Lennox (Hardy Diagnostics, Santa Maria, CA, USA) per liter of water. All the *E. coli* cell colonies were grown on LB plates composed of 25.00 g Lennox LB and 12.00 g agar powder (Alfa Aesar, Haverhill, MA, USA) per liter of water. Depending on the experiment and type of selection needed, kanamycin, arabinose, and/or levofloxacin were added to the LB media and LB plates. An overnight starter culture was made of 8 mL of LB, 8 µL of 50 mg/mL kanamycin, and one bacterial colony combined in a 15 mL conical tube and incubated with shaking at 37 °C.

### 2.2. Growth Curves

A growth curve was created by measuring the OD_600_ of different conditions over 15 h. In total, 20 µL starter culture was diluted into a new conical tube containing 5.5 mL of LB, and this solution was then further incubated with shaking at 37 °C for 2 h to ensure exponential growth phase. Conditions for growth included 28 °C or 37 °C, and 0%, 0.1%, or 0.5% (*w*/*w*) of L-arabinose and/or 25 ng/mL levofloxacin. For each condition, 3 mL of solution was prepared. In a 96-well clear flat-bottom plate (Corning Life Sciences, Corning, NY, USA), 300 µL of solution was dispensed into each well in the column corresponding to a given condition, resulting in eight technical replicates. Using the kinetics mode on a Spectramax i3X (Molecular Devices, San Jose, CA, USA), the instrument was set to record the OD_600_ value every 15 min for 15 h with a 5 s shake before reading. The temperature was set to either 28 or 37 °C.

Growth curve data was plotted in GraphPad Prism 10, where the mean and standard error were calculated. A nonlinear, logistic growth fit of the data provided the rate constants and maximum absorbance values. Significant differences in the logistic growth data were determined using an ordinary one-way analysis of variance (ANOVA) with a post hoc Tukey test.

### 2.3. Crystal Violet Assay

Biofilm was grown with 0.2 g of sterile glass wool (Sigma-Aldrich) in a clear 6-well plate (VWR, Radnor, PA, USA) as described previously [12,18,19]. For the 0% condition, each well contained 5 mL of LB, 150 µL of starter culture, 5 µL of kanamycin (50 mg/mL), and 250 µL of sterile water. For the 0.5% (*w*/*w*) condition, each well contained 5 mL of LB, 150 µL of starter culture, 5 µL of 50 mg/mL kanamycin, and 250 µL of 10% (*w*/*w*) L-arabinose. For 0.1% (*w*/*w*), all conditions were identical except 50 µL 10% (*w*/*w*) L-arabinose was added. The 6-well plates were grown at either 28 °C or 37 °C for 48 h with a media change performed at 24 h. An OD_600_ of the supernatant was measured after 24 and 48 h. After 48 h of growth, the bacterial cultures in each well were discarded and the glass wool was rinsed three times with ultrapure water using a serological pipette. A 0.1% (*w*/*v*) crystal violet solution used to stain the microtiter plate biofilms was made by dissolving 0.0500 g of crystal violet powder (Sigma-Aldrich) in 50 mL of MilliQ ultrapure water. After rinsing, 5 mL of crystal violet was added to each well and left to incubate at room temperature for 15 min. Once stained, the glass wool was rinsed three times with a pH 7.0 sodium phosphate buffer (Sigma-Aldrich and Alfa Aesar). The plates were then left to dry with the lids open overnight in a static 37 °C incubator. A 30% acetic acid mixture used to solubilize the crystal violet was made by mixing 75 mL of acetic acid (Sigma-Aldrich) and 175 mL of ultrapure water. The following day, the crystal violet adhered to the glass wool was solubilized in 30% acetic acid. Stained glass wool was added to a 15 mL conical tube using sterilized forceps along with 6 mL of 30% acetic acid to solubilize the crystal violet adhering to the glass wool. The conical tubes were vortexed and shaken at 200 rpm for 10 min. To measure the amount of crystal violet present in each sample, a 96-well clear plate was used to measure absorbance of a 1:10 dilution of each sample. The OD_590_ was measured using a Spectramax i3X plate reader (Molecular Devices, San Jose, CA, USA) and the OD_590_/OD_600_ ratio was used to quantify biofilm growth. Growth was determined for three or more biological replicates and was plotted in GraphPad Prism 10, where the mean and standard error were calculated. Significant differences were determined using a two-way analysis of variance (ANOVA) with a post hoc Tukey test.

### 2.4. Biofilm Growth on Agar and EPS Harvesting

Biofilm grown for colony biofilm analysis was grown from 15 µL droplets of overnight starter cultures that had been diluted at a ratio of 1:100 in LB and grown for about 2 h until they reached a logarithmic growth phase (OD_600_~0.8). Plates were placed at growth temperature for 48 h prior to analysis. Photos were taken with a ruler in a lightbox with an iPhone Pro 11 and analyzed in ImageJ v.1.53.a. Biofilm grown for analysis of extracellular polymeric substances was grown statically on 10 cm LB-agar plates (VWR) supplemented with 50 µg/mL kanamycin, with or without 0.5% (*w*/*w*) arabinose or 25 ng/mL levofloxacin. Using a sterile wooden inoculating pick, three to five colonies of PHL628 *E. coli* were collected from an agar plate and suspended in 2 mL sterile saline (0.9% *w*/*v* NaCl) in a 15 mL conical tube. The solution was vortexed and adjusted to an OD_600_ of 0.08 to 0.1. A sterile cotton swab was dipped into the tube, excess liquid was removed, and the swab was streaked over the surface of the agar plates with even distribution next to a lit Bunsen burner and allowed to dry. Plates were transferred and incubated at 28 or 37 °C for 48 h.

After 48 h, a sterile 25 cm cell scraper was used to remove the layer of biofilm from the top of the agar; the biofilm was transferred to the microcentrifuge tube using a pipette tip, and the wet mass was recorded and used to normalize for total biofilm growth. Then, 750 µL of 1.5 M NaCl in ultrapure water was added and gently vortexed into a homogeneous slurry. The solution was incubated at room temperature for 5 min, and vortexed every minute. Samples were centrifuged for 10 min at 5000× *g*. For each condition, approximately 350 µL of supernatant was transferred into two new sterile microcentrifuge tubes to be stored at −20 °C. The remaining pellet was left in the centrifuge tube in the 37 °C oven to dry for two or more days. The dry mass of the harvested cells was recorded.

To quantify the protein concentration in the EPS, a bicinchoninic acid (BCA) protein assay was performed (ThermoScientific, Waltham, MA, USA). A stock solution of Bovine Serum Albumin (BSA, ThermoScientific, 2 mg/mL) was used to generate a standard curve spanning concentrations from 25 to 2000 μg/mL in a 1.5 M NaCl solution, including a blank sample (0 μg/mL BSA). Then, 25 µL of each standard and 1:10 dilutions of EPS (stock: 1.5 M NaCl) were pipetted into a clear 96-well plate (Corning, Corning, NY, USA). All EPS unknowns were performed in duplicate. 200 µL of the working reagent was added to all wells and mixed via pipetting. The plate was incubated at 37 °C for 30 min followed by 10 min at room temperature. Absorbance was measured at 562 nm using the Molecular Devices Spectramax i3X microplate spectrophotometer. A standard curve was generated by plotting the blank-corrected absorbance values at 562 nm for each BSA standard against their respective concentrations (μg/mL) and used to calculate protein concentration.

A phenol-sulfuric acid assay was conducted for carbohydrate quantification [20]. In total, 200 µL of the sample was added to 200 µL 5% (*v*/*v*) phenol (Sigma-Aldrich) in ultrapure water and 1.0 mL concentrated sulfuric acid (Flinn Scientific, Batavia, IL, USA) in individual microfuge tubes; the tubes were mixed and incubated at room temperature for 10 min. Samples were then mixed well and placed in a 30 °C water bath for 20 min. Glucose standards were prepared simultaneously. Then, 200 µL of the treated samples were pipetted into a clear 96-well plate (Corning) and the absorbance at 490 nm was measured on a Spectramax i3X plate reader (Molecular Devices).

Protein and carbohydrate concentrations were plotted relative to wet biofilm mass of the collected sample in GraphPad Prism 10, where the mean and standard error were calculated. Significant differences were determined using a two-way analysis of variance (ANOVA) with a post hoc Tukey test.

### 2.5. Confocal Laser Scanning Microscopy (CLSM)

Biofilm cells were grown on glass wool in a manner identical to the method described above in the crystal violet assay. An OD_600_ of the supernatant was measured after 24 and 48 h. After 48 h, a small piece of glass wool approximately 1 inch long was removed for each condition and washed three times with MilliQ ultrapure water in a Petri dish to remove planktonic cells. For the calcofluor white (Sigma-Aldrich) staining, the small piece of washed glass wool was placed on a slide and 1 drop (approximately 20 µL) of 10% KOH (Sigma-Aldrich) followed by 1 drop of calcofluor white were added using a transfer pipette. For the FilmTracer SYPRO Ruby (Invitrogen, Waltham, MA, USA) staining, the sample of glass wool was placed in a 24-well plate (VWR) with 3 mL of FilmTracer SYPRO Ruby stain and incubated in the dark for 30 min. The sample was then gently rinsed once with MilliQ water and placed on a slide. A cover slip was placed on top of each sample. Confocal laser scanning microscopy (CLSM) was employed to analyze the PHL628 biofilm cells and extracellular polymeric substances using a Zeiss LSM 880 inverted confocal microscope (Carl Zeiss, Jena, Germany) equipped with MA-PMT and GaAsP array detectors. Samples were viewed through a 10× objective. FilmTracer SYPRO Ruby-stained samples were imaged by excitation from a 458 nm argon laser, while calcofluor white-stained samples were imaged by excitation from a 405 nm diode array. Fluorescence intensity analysis was completed by imaging five randomly located, identically sized areas on a slide in the same z-plane of greatest fluorescent intensity. Images were analyzed in ImageJ by determining a common threshold of fluorescence using Otsu thresholding and subsequently determining the mean integrated density of fluorescent areas using the “analyze particles” function [21,22,23]. This analysis provided us with comprehensive information about the spread and size of stained biofilm in two dimensions, as well as fluorescence intensity. The integrated density weighted to particle size was plotted in GraphPad Prism 10, where the mean and standard error were calculated. Significant differences were determined using a two-way analysis of variance (ANOVA) with a post hoc Tukey test.

### 2.6. RNA Harvesting

A pH 7.0 0.2 M potassium phosphate (Alfa Aesar) and a pH 7.4 10 mM Tris-HCL (G Biosciences, St. Louis, MO, USA) were prepared for harvesting biofilm from glass wool. Planktonic and biofilm bacterial cells were grown in 500 mL Erlenmeyer flasks with glass wool. Approximately 1 g of glass wool was autoclaved in a 500 mL Erlenmeyer flask loosely covered with aluminum foil. Into each flask, 100 mL LB, 103 µL 50 mg/mL kanamycin, 3 mL starter culture, and 5.4 mL autoclaved MilliQ water or 10% (*w*/*w*) L-arabinose was added. Flasks were incubated in a 28 or 37 °C rotary shaking incubator at 200 rpm. After 24 h, the supernatant in the flask was removed using serological pipettes, replaced with fresh media and returned to the shaking incubator.

Approximately 24 h after the media change and 48 h after the start of the experiment, planktonic cells were harvested from the liquid supernatant and biofilm was collected after detachment from glass wool. A sample of planktonic cells was collected in 15 mL conical tubes and the rest of the media was decanted from the Erlenmeyer flask. The glass wool was gently rinsed three times with 50 mL of 0.2 M potassium phosphate buffer to remove any residual planktonic cells. The glass wool was then transferred into a 250 mL autoclaved Erlenmeyer flask using flame-sterilized tweezers with approximately 8 g of autoclaved glass beads. Then, 10 mL of Tris-HCL buffer was added, and flasks were shaken at 300 rpm in the rotary shaker for 10 min. After shaking, the solution was decanted into 15 mL conical tubes and the optical density at 600 nm (OD_600_) was measured using an Eppendorf BioPhotometer for planktonic and biofilm samples. The Genomics Agilent calculator was used to calculate the volume of each condition needed to harvest approximately 1.2 × 10^9^ cells/mL in sterile microcentrifuge tubes. After centrifuging for 10 min at 8000× *g*, the supernatant was decanted, and cell samples were placed in the −80 °C freezer or processed immediately for RNA purification.

Planktonic and biofilm cell samples were removed from the −80 °C freezer on ice or immediately taken from harvesting to purify. Then, 1 mL of RNAprotect Bacteria Agent (Qiagen, Germantown, MD, USA) was added to cell pellets and vortexed for 5 s, followed by a 5 min incubation at room temperature. Microcentrifuge tubes were centrifuged for 10 min at 5000× *g* followed by decantation of the supernatant. A TE buffer consisting of 30 mM Tris-Cl and 1 mM EDTA at pH 8.0 (Invitrogen) was combined with lysozyme (Sigma-Aldrich) for a final concentration 15 mg/mL lysozyme. This was further combined with 20 mg/mL proteinase K (ThermoFisher) for a final concentration of 3 mg/mL. Then, 115 µL of this TE/lysozyme/proteinase K mixture was added and vortexed for 10 s. Following this addition, the microcentrifuge tubes were incubated at room temperature for 10 min, vortexing for 10 s every two minutes. The Qiagen RNeasy Kit (Qiagen) was used to purify the samples. A ThermoFisher NanoDrop One (ThermoFisher) was used to measure the concentration and purity of samples.

### 2.7. RNA-Seq Data Collection and Analysis

For RNA-seq, samples of RNA were harvested and purified as described above. Three biological replicates were prepared for each experimental condition. At least 5 µg of RNA was added to GenTegra Tubes (GenTegra, LLC, Pleasanton, CA, USA) and diluted to a final volume of 50 µL with RNase-free water (Qiagen). Samples were frozen with liquid nitrogen and lyophilized (VirTis, −55 °C, 20 mTorr, overnight) to remove water. Samples were then stored at room temperature (20–22 °C). Samples were analyzed by Mr. DNA (Molecular Research LP, Shallowater, TX, USA). The lyophilized total RNA was resuspended in 25 µL nuclease free water. The concentration of the RNA was determined using the Qubit^®^ RNA Assay Kit (Life Technologies, Carlsbad, CA, USA) and the RNA integrity number (RIN) was determined; all samples had a RIN greater than 7.1 µg total RNA was used for rRNA removal by using Ribo-Zero Plus rRNA Depletion Kit (Illumina, San Diego, CA, USA). rRNA depleted samples were quantified and used for library preparation using the KAPA mRNA HyperPrep Kits (Roche, Basel, Switzerland) by following the manufacturer’s instructions. Following the library preparation, the final concentration of all the libraries was measured using the Qubit dsDNA HS Assay Kit (Life Technologies), and the average library size was determined using the Agilent 2100 Bioanalyzer (Agilent Technologies, Santa Clara, CA, USA). The libraries were then pooled in equimolar ratios of 0.6 nM, and underwent paired-end sequencing for 300 cycles using the NovaSeq 6000 system (Illumina). The data discussed in this publication have been deposited in NCBI′s Gene Expression Omnibus [24] and are accessible through GEO Series accession number GSE 299716 (https://www.ncbi.nlm.nih.gov/geo/query/acc.cgi?acc=GSE299716, accessed 14 July 2025).

To analyze the RNA sequencing data, a data workflow in R v4.4.1 was used as previously described, with modifications highlighted below [12]. The workflow and source code can be accessed from the Ryno Lab Github RNA-seq Repository (https://github.com/OCRynoLab/ArabinoseRNAseq, accessed on 14 July 2025). Briefly, RNA-Seq data quality control, alignment, quantification, and statistics calculations were conducted with the workflow, using the reference genome and annotation of *E. coli* strain MG1655 from NCBI (genome sequence GenBank ID U00096.3; assembly ID GCA_000005845.2). Adapter sequences were trimmed with scythe v0.994, and low-quality ends were trimmed with sickle v1.33 [25,26]. HISAT2 v2.2.2.1 was used to align reads to the reference genome [27]. Salmon v1.10.2 was used to quantify the expression of transcripts [28]. Using the tx2gene program in the Bioconductor R package v3.19 [29], we created GFF files and compared them to the assembled and annotated *Escherichia coli* str. K-12 substr. MG1655 (Taxonomy ID: 511145, GCA_000005845). Features of interest were listed as a “gene”, with each gene identified by MG1655 locus tag. The resulting GFF file contained 4566 features (4240 protein coding, 147 pseudogene, 71 noncoding RNA, 22 rRNA, and 86 tRNA). For principal component analysis (PCA), the tximport R package v. 1.36.1 was used to import the Salmon data, edgeR was used to calculate normalization factors and log-transformed CPMs, and the *prcomp()* function in R was used to calculate PCA [30,31]. Differentially expressed genes (DEGs) were identified by subjecting raw counts to the DESeq2 package v1.44.0 in R v4.3.2 [32]. The Benjamini and Hochberg False Discovery Rate (FDR) criterion was used to compute *P_adj_* values. An absolute value of log_2_ fold change  >  2 (i.e., a four-fold difference in either direction) and a *P_adj_*  <  0.01 was used as the threshold for selecting DEGs. Venn Diagram of DEGs was created using the eulerr package v. 7.0.2 for R [33]. All data was plotted in GraphPad Prism 10. Linear regression was conducted on comparative analyses and the fit was reported in the respective figures. Gene ontology analyses were conducted on DEGs using PANTHER and EcoCyc [34,35]. Enrichment testing was conducted in PANTHER using the over-representation tool [36]. All data was plotted in GraphPad Prism 10.

### 2.8. RT-PCR/qPCR

qPCR reactions were performed on cDNA prepared from total cellular RNA using the QuantiTect Reverse Transcription Kit (Qiagen). The Rotor-Gene SYBR Green dye (Qiagen), cDNA, and appropriate primers purchased from Integrated DNA Technologies (Appendix A) were used for amplifications (45 cycles of 2 min at 95 °C, 10 s at 95 °C, 30 s at 60 °C) in a Qiagen Rotor-Gene Q instrument. All transcripts were normalized to the housekeeping gene *rrsA* and all measurements were performed in triplicate. Data was analyzed using the ΔΔCt method and error was reported as SEM. Data was plotted in GraphPad Prism 10, where the mean and standard deviation were calculated. Significant differences were determined using a two-way analysis of variance (ANOVA) with a post hoc Tukey test.

## 3. Results

### 3.1. Arabinose and Levofloxacin Influence E. coli Planktonic Cell and Biofilm Growth

When we measured the growth of planktonic PHL628 *E. coli* cultures in rich media using optical density measurements at 600 nm (OD_600_), we observed notable differences between our control condition (0% sugar added) and L-arabinose addition at both experimental temperatures (Figure 1A,B, Table 1). *E. coli* growth rate increased with statistical significance upon the addition of arabinose: for example, we observed 2.5 times the growth rate at 37 °C with 0.5% (*w*/*w*) arabinose compared to the control (*p* < 0.0001) but found depressed maximum growth values compared to the control (*p* < 0.008).

Interestingly, the addition of levofloxacin to a culture did not show a significant difference in the growth rate constant relative to either control for either temperature. We observed either a temperature-dependent depression (28 °C) or enhancement (37 °C) with the combined addition of levofloxacin and arabinose. For both temperatures, levofloxacin decreased the maximum growth of the cultures, and the addition of arabinose to levofloxacin-treated cultures significantly increased the maximum growth compared to levofloxacin alone (Table 1, ‡). Experiments measuring biofilm growth on glass wool revealed that the addition of 0.5% (*w*/*w*) arabinose, 25 ng/mL levofloxacin, or a combination of both significantly increased the amount of biofilm formed compared to control at 37 °C (Figure 1C). At 28 °C, we observed no significant difference in biofilm growth across our experimental conditions.

We examined the radial growth of 48 h colony biofilms on agar substrate containing arabinose, levofloxacin, or a combination at various temperatures (Figure 2). Interesting morphological differences emerged among the conditions; notably, with the addition of arabinose, particularly at 37 °C, we see the prevalence of a thin, broad “outer rim” region with irregular edges of the colony biofilm (Figure 2A). 25 ng/mL levofloxacin significantly disrupted growth of the biofilm colony, resulting in small microcolonies formed in the area where the original drop of *E. coli* solution was added at 28 °C, and a small, dense colony at 37 °C (Figure 2A,B).

Arabinose addition in the presence of levofloxacin recovered biofilm colony growth at both temperatures (Figure 2A 25 ng/mL Lev vs. 0.5% Ara + 25 ng/mL Lev). Colony growth with both arabinose and levofloxacin added was not significantly different from the control but smaller than the observed growth with arabinose alone (Figure 2B). At both experimental temperatures, the outer rim of these biofilm colonies grew substantially more with the addition of arabinose, even when levofloxacin was present (Figure 2C). Remarkably, the addition of arabinose does not change the size of the center portion of the biofilm colony, but the addition of levofloxacin significantly decreases the center colony size (Appendix A). Through these experiments, we see that arabinose has a significant effect on biofilm colony formation, promoting radial growth and survival during antibiotic insult.

### 3.2. Arabinose Addition Changes Biofilm EPS Composition

After determining that arabinose promoted PHL628 biofilm growth on both glass wool and agar, we were interested in examining if the extracellular polymeric substance (EPS) composition of the biofilms changed with regard to protein or carbohydrate concentration. We examined EPS harvested from lawns of biofilm grown on agar using colorimetric assays and biofilm colonies grown on glass wool using confocal microscopy [12,37]. We recorded the mass of our biofilms prior to EPS extraction (wet mass, collection of EPS and biofilm cells) and the mass after EPS extraction and drying (dry mass, dried cells and non-soluble debris) (Figure 3A, Appendix A). Examination of the wet mass reveals that the addition of arabinose does not induce a statistically significant change in growth at either 28 or 37 °C when we grow biofilms as a lawn on agar with arabinose as compared to the control (Figure 3A).

As expected, we see more absolute biofilm for arabinose samples at 28 °C compared to 37 °C (*p* = 0.0204), likely due to better development and maintenance of curli at growth temperatures below 30 °C [12,17,38]. Like the colony biofilm experiments, the addition of levofloxacin at a concentration of 25 ng/mL to our agar led to spotty biofilm colony growth and significantly less biomass (Figure 3A, Appendix A). Interestingly, the addition of arabinose increased biofilm mass in the presence of levofloxacin in a statistically significant manner at both experimental temperatures compared to levofloxacin alone. These differences from our biofilm results on glass wool are not completely unexpected, as we have observed that different substrates promote different biofilm growth patterns [37].

After harvesting our EPS from the biofilm grown on agar, we normalize the concentration of the EPS biomolecule to the amount of biofilm collected to account for differences in biofilm growth. At 28 °C, we observe similar independent influences of arabinose and levofloxacin for both normalized protein and carbohydrate concentrations: their addition increases the amount of protein or carbohydrate in the EPS, potentially from inducing cell death and lysis in the case of levofloxacin, and increasing the concentration of soluble biomolecules per amount of biofilm in the samples. Concentrations of these biomolecules were not statistically different from control when levofloxacin and arabinose were added simultaneously. At 37 °C, we saw no significant changes in EPS protein concentration for biofilms grown on agar substrate, but we observed an increase in EPS carbohydrate with the addition of L-arabinose (Figure 3B,C). The combined presence of arabinose and levofloxacin depressed the amount of normalized carbohydrate concentration relative to control. On glass wool, we see fewer significant changes in EPS composition with the addition of arabinose or levofloxacin at either temperature studied, as determined by confocal microscopy (Appendix A). We note that there was no substantive change in EPS protein concentration as determined by SyproRuby dye quantification when arabinose was present, but we did observe an increase in EPS protein with the addition of levofloxacin at 37 °C.

### 3.3. Examining Changes to the E. coli Transcriptome with Arabinose Addition

At this juncture, our experiments revealed significant changes to bacterial growth and EPS composition with the addition of arabinose, but did not provide information about which internal signaling and metabolic pathways and cellular systems were changing to cause these gross phenotypic changes. We mapped the transcriptome of eight experimental conditions: cells grown with or without 0.5% (*w*/*w*) arabinose, in the planktonic or biofilm state, at either 28 or 37 °C (Figure 4, Appendix A).

Principal component analysis (PCA) of our samples demonstrated clustering of biological replicates of the same condition, as well as dominant clustering by temperature, rather than by biofilm/planktonic or arabinose condition (Figure 4A). Our datasets revealed numerous differentially expressed genes (DEGs, Fold Change > 4, FDR < 0.01, (Appendix A)). Analysis of differentially expressed genes upon arabinose addition revealed surprising areas of rich overlap (e.g., 28 °C biofilm samples and 37 °C planktonic samples), as well as expected areas of overlap with temperature-matched (e.g., 28 °C planktonic and biofilm) or condition-matched (e.g., biofilm) samples.

There were 28 genes that were differentially expressed in all conditions. This overlapping set comprises genes consistently upregulated across all conditions, such as those involved in arabinose catabolism (*araA*, *araB*, *araD*), as well as genes consistently downregulated including those required for glutamine and arginine catabolism (*astA*, *astB*, *astD*, *astE*, *gltB*) (Appendix A). We confirmed our RNAseq results by conducting RT-qPCR on the genes *araE*, *csgA*, and *fimA*, which replicated the trends we observed with our RNAseq dataset (Appendix A). Based on our PCA results and trends observed among overlapping DEGs across all experimental conditions, we further examined each temperature set independently to understand what genes were being differentially expressed in planktonic versus biofilm cells at a given temperature.

### 3.4. Influence of Arabinose on Gene Expression at 37 °C

When focusing on data collected at 37 °C, we observe an overlap of 200 genes that are differentially expressed in both planktonic (N = 1216 DEGs) and biofilm (N = 400 DEGs) samples (Figure 5A). We observed 63 genes that were upregulated in both planktonic and biofilm samples (upper right quadrant of Figure 5A). When subjecting these genes to gene ontology biological process overrepresentation analysis (ORA), we observed the expected significant enrichment of L-arabinose catabolic processes (GO:0046373, GO:0019572, GO:0019569), and, gratifyingly, the enrichment of biofilm-related processes (GO:0044010, GO:0098743, 0042710, GO:0098630) (Appendix A). When we further explore the genes mapped to these biofilm processes, we see several toxin–antitoxin systems represented (*msqR*-*msqA*, *relE*-*relB*, *and yhaV*-*prlF*), as well as the biofilm dispersal mediator *bdcA*, and the cell envelope stress-responsive sigma factor *rpoE*. The upregulated subset also overrepresented the biological processes of DNA-templated transcription regulation (GO:0006355, GO:2001141, GO:0051252), which has several genes that overlap with biofilm processes, such as the toxin-antitoxin systems, but also transcriptional regulators like *soxR* and *stpA*.

We observed 122 DEGs downregulated in both planktonic and biofilm samples exposed to arabinose at 37 °C (lower left quadrant of Figure 5A). We found that transport processes, specifically those related to nitrogen compound transport (GO:0071705), glycine import (GO:1903804), tripeptide and oligopeptide transport (GO:0042939 and GO:0035672), arginine catabolism (GO:0006527, GO:0019544, GO:0019545), and certain biosynthetic processes (GO:0009058, GO:0019540, GO:0033068, GO:0009239, and GO:0030639) were overrepresented in our downregulated dataset (Appendix A). Homing in on transport processes, we noted that the downregulated genes included several ATP-binding cassette transporters (*proVWX*, *oppCDF*, and *dppD*), protein exporters (*secDF* and *yidC*), proton-dependent oligopeptide transporters (*dtpAD*), and spermidine ATP-dependent transporters (*potBCD*), implicating impact on the transport of several types of biomolecules and small molecules, which could be contributing to changes in the EPS.

We found 15 DEGs upregulated in planktonic but downregulated in biofilm samples (upper left quadrant, Figure 5A). Overrepresentation analysis revealed that lipopolysaccharide core region biosynthetic processes (GO:0046401, GO:0009244) attributed to the *waaBJOSUZ* and the *rfbCX* operons were upregulated in this cohort (Appendix A). Direct examination of the DEG list also showed genes in the glutamate-dependent GAD system (*gadBCE*) in this quadrant, which are components of the principal acid resistance system, pH homeostasis, and multidrug efflux. We found no DEGs that were downregulated in planktonic cells but upregulated in biofilm.

### 3.5. Influence of Arabinose on Gene Expression at 28 °C

At 28 °C, we observe an overlap of 467 genes that are differentially expressed in both planktonic (N = 886 DEGs) and biofilm (N = 1514 DEGs) samples (Figure 5B). We observed 130 genes that were upregulated in both planktonic and biofilm samples (upper right quadrant of Figure 5B). When subjecting these genes to gene ontology biological process ORA, in addition to the expected arabinose catabolic processes, we observed overrepresentation of genes related to ammonia assimilation (GO:0019676), amino acid transport (GO:0098712, GO:0140009), amino acid biosynthetic processes (GO:0097054, GO:0000105), and de novo nucleobase biosynthetic processes (GO:0006207, GO:0044205) (Appendix A).

Of the 258 genes that are downregulated with arabinose exposure in both biofilm and planktonic samples at 28 °C, we found the genes that code for the structural components of curli (*csgA* and *csgB*) were both downregulated. With biological process ORA, we found that arginine catabolic processes (GO:0019545, GO:0019544) are overrepresented within this data subset, as well as certain stress responses (GO:0009271, GO:0006950). Additionally, we found that for the 68 genes that are upregulated in biofilm but downregulated in planktonic samples at 28 °C, there are overrepresented processes related to the aerobic electron transport chain (GO:0019646, GO:0015988, GO:0009060, GO:0022904), peptide transport (GO:0035442, GO:0042938) and glycine catabolism (GO:0019464, GO:0006546). Of the 11 genes that are differentially downregulated in biofilm but upregulated in planktonic samples at 28 °C, we observe genes that play a role in stress responses and protein folding like *ccp*, *yhcN*, *cbpM*, and *uspC*.

### 3.6. Arabinose Changes Expression of Genes RELATED to Extracellular Structures, Efflux, Transporters, and Antibiotic Response

We observed interesting patterns of gene expression for those genes within the pilus GO category (GO:0009289) that appeared to correlate with the state of the samples (planktonic versus biofilm) (Figure 6A). For planktonic cells, regardless of temperature, we see more upregulation of pilus genes with the notable exception of the fimbriae genes (*fimAFGHI*) at 37 °C. For biofilm cells, we observe the most DEGs in this subcategory at 28 °C, and, except for the fimbriae genes just mentioned, we observed significant downregulation of pilus genes.

Since our biofilm colony growth experiments showed enhanced growth in the presence of arabinose when bacteria were subjected to sub-MIC levofloxacin exposure compared to control, we examined genes in the antibiotic response biological process GO (GO:0044667) to see if the addition of arabinose at either experimental temperature changed genes known to participate in canonical *E. coli* responses to antibiotics (Figure 6B, Appendix A). Notable trends include the differential upregulation of ribosomal proteins in 28 °C biofilm and downregulation of those same proteins for biofilm grown at 37 °C and correlating up and downregulation of transcription/translation related genes for those samples.

We see the biggest differences in expression for transporters at 37 °C, though the response is mixed and dependent upon transporter type. When we looked at transporter genes and categorized them based on their cargo and/or function, we noted that there are significant differences in gene expression based not only on temperature but also on physical state (Appendix A). For example, for planktonic samples grown at 37 °C, we observe the upregulation of the cryptic type II secretion system, certain efflux pumps and amino acid and peptide transport, carbohydrate transport, and activation of the electron transport chain. For these same samples, porin and stress responsive gene expression is downregulated, as is lipid and vitamin transport. Biofilm samples grown at 28 °C have a notable increase in expression of amino acid and peptide transporters, as do porins and osmotic stress response genes (e.g., *proVWX*, *ompC*, and *osmF*). Interestingly, for biofilm samples grown at 37 °C, these same genes are downregulated except for *osmF*.

Efflux pumps, some of which were represented in the antibiotic response GO category as well as in the transporters category, are of particular interest as they could promote efflux of toxic metabolic intermediates that may accumulate with the addition of arabinose or antibiotics [39]. We curated a list of efflux pumps from Ecocyc and found that for planktonic cells at 28 °C, the *mdt* (multi drug transporter) efflux pump DEGs were largely upregulated, with the notable exception of *mdfA*, which was downregulated in both planktonic and biofilm conditions at 28 °C and in planktonic conditions at 37 °C (Appendix A). At 37 °C, we note a mix of up and downregulation of efflux pump genes: we see that *mdtM* and *arsB*, which export unconjugated bile salts and arsenate, respectively, are upregulated in planktonic cells, while *setB*, a sugar exporter, and *yojI*, an ABC-type exporter, are downregulated. Collectively, our analysis of RNA-seq data reveals that arabinose changes gene expression in both a temperature- and physical state-dependent manner.

## 4. Discussion

Our examination of L-arabinose’s influence on *E. coli* growth, biofilm formation and composition, and expressed genes revealed nuanced responses dependent on both growth temperature and physical state of the bacteria. Moreover, we discovered that the presence of arabinose enhanced survival of *E. coli* cells in the biofilm state when challenged with levofloxacin, a phenomenon we had previously only observed for cells in the planktonic state [16].

The study of L-arabinose metabolism in *E. coli* has been largely focused on the hierarchy of non-glucose sugars and carbon catabolite repression [7,40,41], which places L-arabinose as a favored carbon source only lower in the hierarchy than the disaccharide lactose. We observe here a significant increase in the planktonic growth rate and a decrease in maximal growth when L-arabinose is added at concentrations above saturating conditions (>0.2% (*w*/*w*)) in media that contains additional carbon sources (Luria–Bertani) [40]. Therefore, it is not surprising that we observe enhanced growth with the addition of this favored carbon source. Additionally, there is considerable evidence that the phosphorylated metabolic intermediates of carbohydrates lead to cellular toxicity in many different types of microorganisms, including *E. coli* [39]. In the L-arabinose metabolic pathway, the sugar ribulose-5-phosphate can accumulate and lead to lytic bulge and cell lysis [42,43]. While the underlying cause of the ribulose-5-phosphate toxicity was not confirmed, it is thought to be linked to UDP-galactose accumulation, which, in turn, prevents the formation of UDP-N-acetylglucosamine, a critical component of the peptidoglycan structure [39]. In our study, we may be observing sugar-phosphate toxicity, especially with saturating concentrations of L-arabinose, that leads to overall diminished total growth.

Several labs have sought to quantify the spatiotemporal changes in nutrient access and metabolism of growing colony biofilms on agar substrate, including significant differences in gene expression, oxic pathways, and viability against toxic insult [11,44,45]. Additionally, nutrient exchange exists within the biofilm, leading to subpopulations of cells responding to varied levels of specific nutrients. In *E. coli* there are several reports documenting the movement of amino acids from the interior of the biofilm to the periphery and exchange of lipids from the periphery to the interior [11,46]. Recently, Zhang et al. determined that glucose is differentially consumed within a biofilm, where cells at the periphery maintain persistent glucose consumption until approximately 225 +/− 7 µm from the edge of the community [47]. The morphological changes we quantified here could signify discrete portions of the biofilm colony that are metabolically and transcriptomically unique.

Others have demonstrated that the presence of glucose influences the sensitivity of bacteria to specific classes of antibiotics [47,48], but the only study, to our knowledge, of arabinose-specific effects in combination with antibiotics has been conducted by our laboratory [16]. Bacteria susceptible to a specific antibiotic will grow less efficiently when there are low concentrations of that antibiotic present in liquid culture, though there is mixed evidence as to whether sub-minimum inhibitory concentrations of antibiotics lead to increased or reduced biofilm growth, depending on the class of antibiotic, species and strain of bacteria, and method used to quantify biofilm growth [49,50,51]. Metabolically active bacteria, in contrast to dormant phenotypes, are more susceptible to antibiotics for many reasons [52]. For example, bactericidal activity is diminished with decreased cellular respiration and can be further negatively influenced with genetically induced decoupling of ATP synthase from the electron transport chain [53]. Small molecules that enhance bacterial metabolism are being explored as clinically relevant adjuvants [54,55]. Bactericidal antibiotics can increase the cellular demand for ATP and generate futile cycles that waste metabolic resources and lead to reactive oxygen species generation [53]. We are surprised, therefore, that the addition of arabinose, a well-metabolized sugar in *E. coli*, increases the minimum inhibitory concentration and enhances bacterial survival in the biofilm state for the antibiotic levofloxacin [16]. This suggests that the influence of arabinose on our system extends beyond its metabolism and use as an energy source.

Compared to the planktonic state, bacteria in the biofilm state are notably more resistant to antibiotic insult, and exposure of *E. coli* to sub-inhibitory concentrations of antibiotics can lead to enhanced biofilm formation and enhanced resilience against antibiotic treatment [56,57,58]. Expression of genes that can promote adoption of the biofilm state include extracellular structures like curli and fimbriae [2]. We observe the upregulation of the structural type I fimbriae gene *fimA* and the *fimF* and *fimG* genes, the products of which act as adaptors between the fimbrial rod and the FimH tip at lower growth temperatures, which are overexpressed in uropathogenic *E. coli* isolates that demonstrate resistance to multiple classes of antibiotics [59]. Additionally, we observe upregulation of the cryptic usher fimbriae system *yadCKLM*, demonstrated to enhance biofilm formation when constitutively expressed in K-12 strains, in our planktonic samples treated with arabinose [60].

We observe that many efflux systems in both our biofilm and planktonic samples are upregulated with L-arabinose, which could explain why L-arabinose confers enhanced survival with levofloxacin treatment in our system. Our transcriptomics data showed that efflux transporters of arabinose, like *ydeA*, are upregulated [61]; *ydeA* overexpression in *E. coli* confers resistance to the sesquiterpenoid natural product xanthorrhizol, which has antibiotic activity [62]. Additionally, many multidrug transporter pumps were upregulated in planktonic samples with arabinose, including the major facilitator superfamily (MFS) genes *emrK*, *emrY*, *mdtH*, and *mdtM* and membrane fusion protein (MFP) genes *mdtE* and *mdtN*. We also observed upregulation of resistance-nodulation-division (RND) pump genes *mdtE* and *mdtF*. Any of these genes could be contributing to reduced sensitivity of *E. coli* to levofloxacin while in the presence of arabinose by increasing levofloxacin efflux. While there is no immediate direct connection between arabinose and the expression of these genes, *araC* gene expression was upregulated nearly 3-fold in clinical isolates of fluoroquinolone-resistant *E. coli* [63]. Others demonstrated the importance of the efflux components *emrKY* and *mdtH* on *E. coli* sensitivity to other fluoroquinolones like ofloxacin and ciprofloxacin [64,65], suggesting a putative avenue of exploration for the mechanism by which arabinose promotes survival with levofloxacin treatment.

## 5. Conclusions

In summary, our findings demonstrate that environmental sugars such as L-arabinose can significantly influence both the growth dynamics and antibiotic susceptibility of *Escherichia coli* in planktonic and biofilm states. L-arabinose enhanced biofilm formation at 37 °C and diminished levofloxacin′s efficacy, promoting growth under sub-MIC antibiotic concentrations. Transcriptomic analyses revealed that these effects are both temperature- and growth-state-dependent. L-arabinose modulates the expression of key genes involved in antibiotic resistance, including efflux pumps, transporters, and biofilm-associated structures such as pili and curli. These results demonstrate a previously unrecognized role for L-arabinose in shaping antibiotic response in *E. coli* and suggest that metabolic context should be carefully considered in efforts to combat biofilm-associated infections and in the development of the next generations of therapeutics.

## Figures and Tables

**Figure 1 microorganisms-13-01665-f001:**
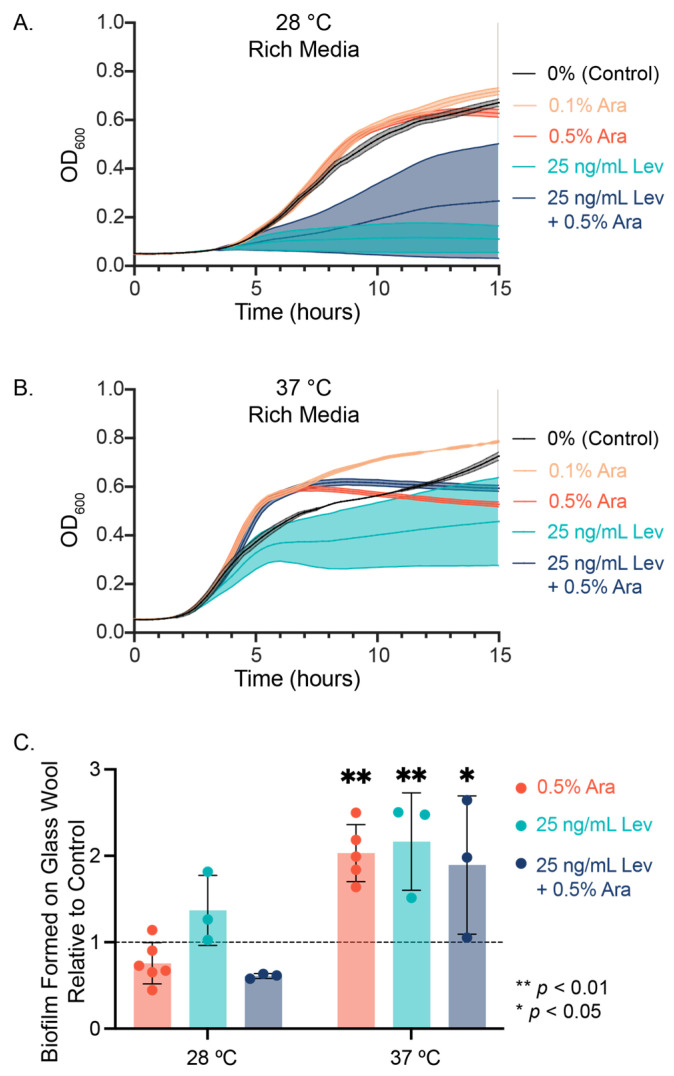
*E. coli* growth with arabinose and levofloxacin. Swimming cell growth was measured at (**A**) 28 °C and (**B**) 37 °C at 600 nm over 15 h. N = 3. (**C**) Biofilm was grown on glass wool over the course of 48 h and stained using crystal violet. Significance was determined using a two-way ANOVA with post hoc Tukey test and reported as ** *p* < 0.01 and * *p* < 0.05 compared to 0% sugar added.

**Figure 2 microorganisms-13-01665-f002:**
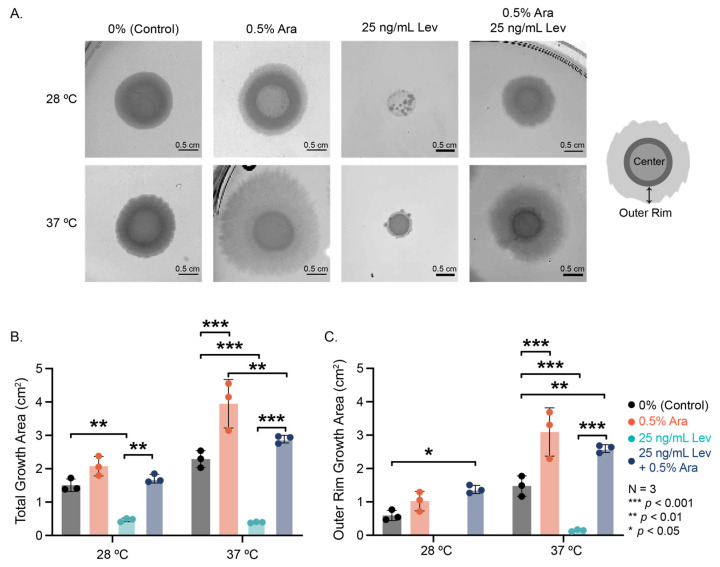
Biofilm colony growth and morphology. (**A**) Representative images of biofilm colonies grown for 48 h at varied experimental conditions. Cartoon depicting our definition of the center of the biofilm colony versus the outer rim. (**B**) Total area measurements and (**C**) outer rim measurements of the biofilm colonies. Note that no outer rim could be determined for levofloxacin biofilm colonies at 28 °C. Significance was determined using a two-way ANOVA with a post hoc Tukey test.

**Figure 3 microorganisms-13-01665-f003:**
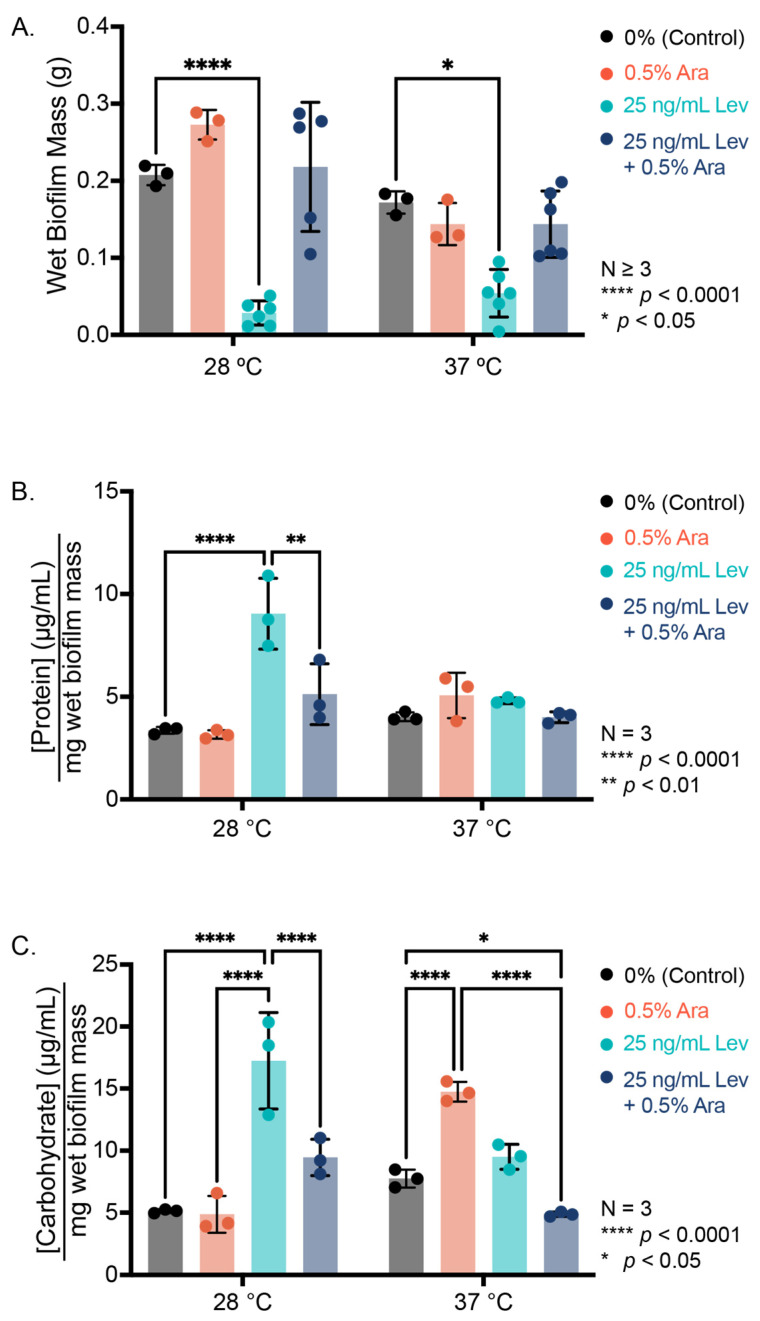
Biofilm growth on agar. (**A**) Wet mass of biofilm formed after 48 h growth at indicated temperature. (**B**) Protein concentration of the EPS determined by bicinchoninic acid assay and normalized to wet biofilm mass. (**C**) Carbohydrate concentration of the EPS determined by phenol-sulfuric acid assay and normalized to wet biofilm mass. Statistical significance was determined by a two-way ANOVA with post hoc Tukey test.

**Figure 4 microorganisms-13-01665-f004:**
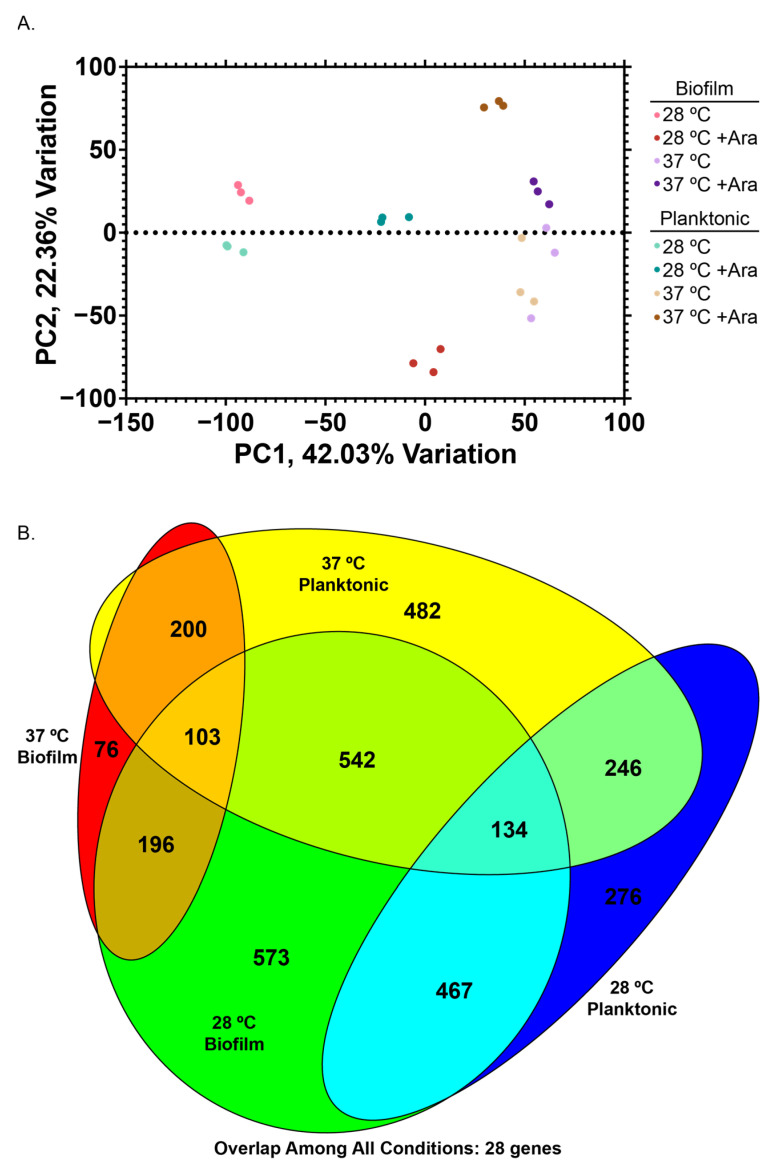
RNA-seq transcriptomics. (**A**) Principal component analysis of eight experimental conditions with three biological replicates each (24 samples). (**B**) Venn diagram highlighting overlap of significantly differentially expressed genes (FC > 4, false discovery rate (FDR) < 0.01) for different conditions, scaled to overlap size.

**Figure 5 microorganisms-13-01665-f005:**
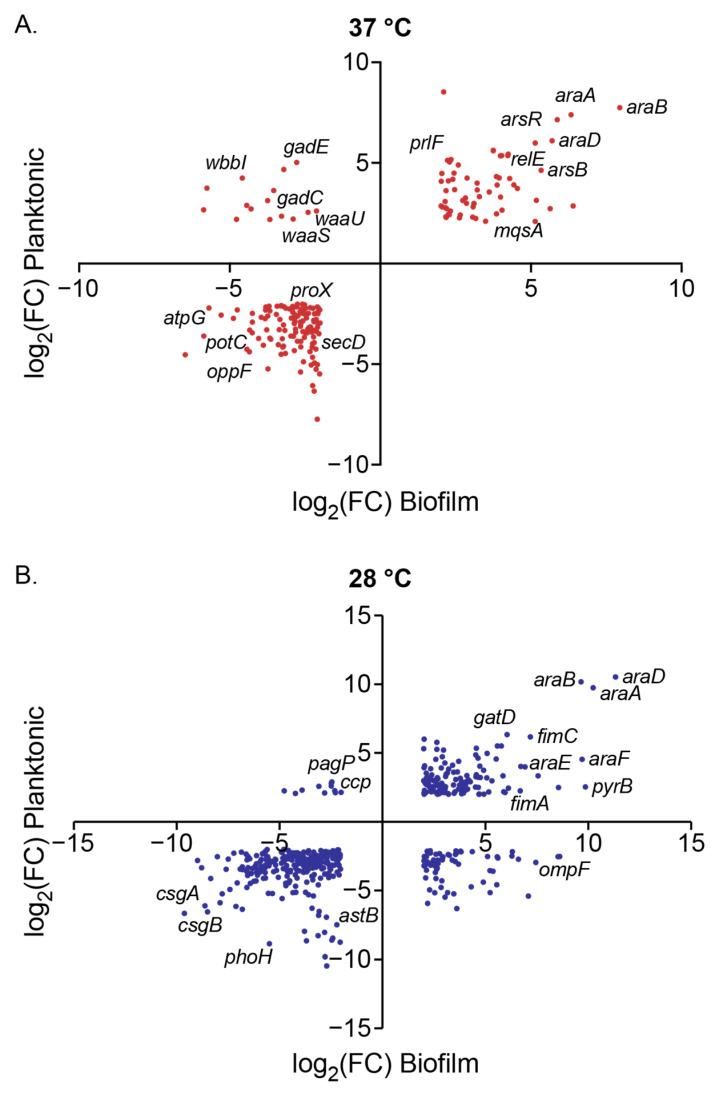
Scatter plots of log_2_ (Fold Change) values of DEGs for planktonic versus biofilm conditions in the presence of 0.5% (*w*/*w*) arabinose compared to control at (**A**) 37 °C and (**B**) 28 °C. Each point represents a DEG (FC > 4 and FDR < 0.01), notable genes are indicated by name.

**Figure 6 microorganisms-13-01665-f006:**
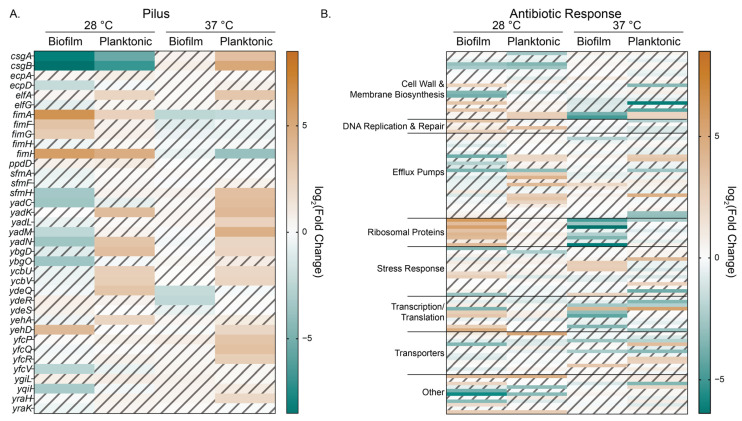
Heat maps of log_2_ (Fold Change) values for genes found in the (**A**) pilus (GO:0009289) and (**B**) antibiotic response (GO:0044667) gene ontology biological process categories. Hashed and faded cells represent genes that did not have significance as determined by our DESeq2 differential expression analysis cutoffs of log_2_(FC) > 2 and *p* < 0.01.

**Table 1 microorganisms-13-01665-t001:** Bacterial growth data (N ≥ 3) presented in Figure 1 were fit to a logistic growth curve to extract the predicted rate constant (k), projected maximum growth (Y_M_), and their respective standard errors using GraphPad Prism 10. Significance was determined using a one-way ANOVA with a post hoc Tukey test.

	Rate Constant (h^−1^)		Maximum Growth (Abs at 600 nm)	
	28 °C	37 °C	28 °C	37 °C
0%	0.500 ± 0.007	0.54 ± 0.02	0.672 ± 0.004	0.639 ± 0.004
0.1% (*w*/*w*) Ara	0.552 ± 0.009	0.674 ± 0.02	0.715 ± 0.004	0.740 ± 0.003 *
0.5% (*w*/*w*) Ara	0.614 ± 0.010 ^§^	1.34 ± 0.04 *^,§^	0.649 ± 0.003	0.567 ± 0.002 *^,§^
25 ng/mL Lev	0.386 ± 0.09	0.74 ± 0.09	0.118 ± 0.005 *	0.423 ± 0.01 *
25 ng/mL Lev + 0.5% (*w*/*w*) Ara	0.24 ± 0.07 *^,‡^	1.12 ± 0.02 *^,†,‡^	0.34 ± 0.09 *^,†,‡^	0.613 ± 0.002 ^†,‡^

* *p* < 0.05 comparing condition to 0%, § *p* < 0.05 comparing 0.1% (*w*/*w*) to 0.5% (*w*/*w*), † *p* < 0.05 comparing levofloxacin to levofloxacin *+* 0.5% (*w*/*w*) arabinose, and ‡ *p* < 0.05 comparing 0.5% (*w*/*w*) arabinose to levofloxacin *+* 0.5% (*w*/*w*) arabinose.

## Data Availability

Workflow and source code can be accessed from the Ryno Lab Github RNA-seq Repository (https://github.com/OCRynoLab/ArabinoseRNAseq, accessed 17 June 2025). The data discussed in this publication have been deposited in NCBI’s Gene Expression Omnibus and are accessible through GEO Series accession number GSE274311 (https://www.ncbi.nlm.nih.gov/geo/query/acc.cgi?acc=GSE299716, accessed 17 June 2025).

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
