# Peer review of "L-Arabinose Alters the E. coli Transcriptome to Favor Biofilm Growth and Enhances Survival During Fluoroquinolone Stress"

_microorganisms, 2025, doi:10.3390/microorganisms13071665_

Round 1
Reviewer 1 Report
Comments and Suggestions for Authors
The publication by Austin and colleagues presents an experimental research using several omics approaches to assess the responses of biofilm and planktonic bacteria to varying temperatures, L-arabinose concentrations, and the presence of levofloxacin. Managing several factors and experimental settings is undoubtedly challenging, therefore I commend you for your excellent effort. Nonetheless, the material has formatting problems and lacks clarity, particularly with the explanation of the "phenotypic" procedures used (e.g., number of replicates, adherence to standard protocols, and replication details). Attend to the quality of the English language. Please refer to the attached version to see my comments and suggestions

Major errors highlighted in the manuscript
Author Response
We thank the reviewer for their meticulous read of our manuscript, and of their attention to the Materials and Methods section. We referred to the attached pdf document and the suggestions and comments were considered in the completion of the updated submission. We attempt to also indicate them below:
Abstract changes:
Lines 11-17 corrected text now reads: “Here, we investigated the growth and composition of Escherichia coli in both planktonic and biofilm states in the presence of L-arabinose, with and without exposure to the fluoroquinolone antibiotic levofloxacin, at two temperatures: 28 and 37 °C. At both temperatures, L-arabinose increased the growth rate of planktonic E. coli but resulted in reduced total growth; concurrently, it enhanced biofilm growth at 37 °C. L-arabinose reduced the efficacy of levofloxacin and promoted growth in sub-minimum inhibitory concentrations.”
Introduction changes:
Line 33: Changed “illnesses” to “diseases.”
Line 56 – 58 corrected text now reads: “Carbon sources are imported and broken down to supply the bacterial cell with amino acids and adenosine triphosphate (ATP) for various biological processes.”
Line 64: Removed “systems.”
Line 65: Changed “utilization” to “used.”
Methods changes:
Line 102/103 reviewer comment: “Materials and methods section, Bacterial stains selection and cultivation subsection, Line 103: which is the origin of the E. coli strain you used? (clinical of field)”
This strain is a derivative of the K-12 MG1655 strain isolated in Philip LeJeune’s laboratory in 1998. While cited in other sections (e.g. Introduction, Line 92: “Here, we examined the independent influence of L-arabinose at different growth temperatures on the biofilm-forming E. coli strain PHL628, which is derived from the parent strain K-12 MG1655 and has a mutation in the ompR gene that leads to constitutive activation and biofilm formation [17].”, we added the citation here in the Methods, too.
Line 104 reviewer comment: “Line 104: it is not clear to me what are you referring to when you say "and 10% (w/w) stock solutions". Stock solutions of what?”
We apologize for the typo and have corrected the line to read: “10% (w/w) L-arabinose (Sigma-Aldrich) stock solutions were made by dissolving 1.00 g of L-(+)-arabinose in 9.00 mL of ultrapure water and sterile filtering the solution using a 0.2 µm polyethersulfone membrane syringe filter (VWR, Radnor, PA, USA).”
Line 109 corrected text now reads: “2 mg/mL levofloxacin (Sigma-Aldrich) stock solution in ultrapure water was prepared and then diluted to 20 µg/mL and sterile filtered.”
Line 114-117 corrected text now reads: “An overnight starter culture was made of 8 mL of LB, 8 µL of 50 mg/mL kanamycin, and one bacterial colony combined in a 15 mL conical tube and incubated with shaking at 37 °C.”
Reviewer Comment: “Materials and methods section, growth curves subsection; Lines 120-130: this paragraph is hard to read and understand. Please refine it using synonims to avoid redundancy. State cleary at the beginnig the different conditions used. Have you used an in-house protocol? or is derived from the literature?”
We apologize for the lack of clarity in our language. This protocol is what we use in house, though it is a typical microbiology growth procedure used for many strains under different conditions. For our case, we are referring to the presence or absence of arabinose and levofloxacin and our two growth temperatures. These conditions are elaborated in Lines 122-123. We removed the paragraph break, which we believe caused a too great a disconnect between the paragraphs.
Line 118-121 corrected text now reads: “A growth curve was created by measuring the OD600 of different conditions over 15 h. 20 µL starter culture was diluted into a new conical tube containing 5.5 mL of LB, and this solution was then further incubated with shaking at 37 °C for 2 hours to ensure exponential growth phase.”
Line 124: “Delivered” was changed to “dispensed.”
Reviewer Comment: “ Line 134: avoid to cite Table 1 here (it collects results not methods), otherwise it is supposed that you'll insert immediately after the first citation.”
We thank the reviewer for noticing this oversight and have removed the reference to Table 2. The text now reads: “A nonlinear, logistic growth fit of the data provided the rate constants and maximum absorbance values.”
Reviewer Comment: “Crystal violet assay subsection; Line 139: the 0% condition can be assumed as negative control? In your method you did not include a negative control as well as a positive control. How many replicates per condition have you tested?
Line 146: it is not clear is the glass wool was retained or not, you simply state that it was rinsed 3 times. Why have you chosen the glass wool?”
Thank you for your questions. The 0% condition is not the negative control—the negative control is the experiment completed without the addition of bacteria (i.e., how much crystal violet dye adheres to the glass wool and is solubilized with acetic acid). Glass wool provides a hydrophilic surface like a glass slide with considerable surface area and sites of nucleation where biofilms grow. We have published this protocol previously (Ref 12. Hantus et al. 2024) and are using it identically. We adopted this protocol from this paper (now Ref 18): (Hayette Benamara, Christophe Rihouey, Thierry Jouenne, Stéphane Alexandre, Impact of the biofilm mode of growth on the inner membrane phospholipid composition and lipid domains in Pseudomonas aeruginosa. Biochimica et Biophysica Acta (BBA) - Biomembranes,
Volume 1808, Issue 1, 2011, Pages 98-105, ISSN 0005-2736, https://doi.org/10.1016/j.bbamem.2010.09.004.
(https://www.sciencedirect.com/science/article/pii/S0005273610003196). We now also cite this paper here on Line 136-137: “Biofilm was grown with 0.2 g of sterile glass wool (Sigma-Aldrich) in a clear 6-well plate (VWR, Radnor, PA, USA) as described previously [12,18,19].”
To address the replicates (we completed at least three biological replicates for each glass wool condition), we now state on Line 159: “Growth was determined for three or more biological replicates and was plotted in GraphPad Prism 10, where the mean and standard error were calculated. Significant differences were determined using a two-way analysis of variance (ANOVA) with a post-hoc Tukey test.”
Line 146 updated: “Once stained, the glass wool was rinsed three times with a pH 7.0 sodium phosphate buffer (Sigma Aldrich and Alfa Aesar).”
Line 137 corrected text now reads: “…, 5 µL of kanamycin (50 mg/mL), and 250 µL of sterile water.”
Line 153 corrected text now reads: “open” instead of “ajar.”
Reviewer comment: “Crystal violet assay subsection; Line 148: in a typical protocol for biofilm staining, you shold first of all fix the bacteria using ethanol or methanol and then stain with the desired dye. Have you use an in-house protocol or derived from the literature?”
Thank you for your comment. We do not fix the bacteria first but move directly from washing to staining. After staining, the wool is dried in an incubator, as indicated in the text, which preserves the bacteria’s structure sufficiently for the liberation of the crystal violet dye. Our protocol is a modified version of the traditional 96-well assay developed by O’Toole for P. aeruginosa and other gram-negative bacteria (Ref 19).
Reviewer comment: “Line 156: why have you moved the glass wool? your protocol was entirely conducted in 6 wells microtiter plates, the solubilizing solution could be kept in the well and underwent slow shake to ensure the ri-solubilization of the dye. Line 160: why the absorbace was measured with an initial 1:10 dilution?”
The glass wool is moved in order for the solubilization process to be most effective and reproducible. We cannot shake a 6-well plate (not microtiter, that designation is usually used for 96-well plates or others with lower volumes and more wells) vigorously enough to achieve reproducible solubilization, therefore we move the wool to a conical tube. The absorbance is measured at a 1:10 dilution because the solubilization results in a very concentrated solution of dye, which needs to be diluted in order for the absorbance to be measured effectively.
Reviewer Comment: “Line 177: 0% agar plates??”
We apologize for the typo and have corrected the text in Lines 173-176 to: “A sterile cotton swab was dipped into the tube, excess liquid was removed, and the swab was streaked over the surface of the agar plates with even distribution next to a lit Bunsen burner and allowed to dry.”
Line 178 corrected text now reads: “After 48 h, a sterile 25 cm cell scraper was used to remove the layer of biofilm…”
Line 188-190 corrected text now reads: “A stock solution of Bovine Serum Albumin (BSA, ThermoScientific, 2 mg/mL) was used to generate a standard curve spanning concentrations from 25 to 2000 μg/mL in a 1.5 M NaCl solution, including a blank sample (0 μg/mL BSA).”
Line 211 corrected text now reads: “2.5 Confocal Laser Scanning Microscopy (CLSM)”
Reviewer Comment: “Line 246: why 30 minutes of autoclave?”
This was the time we used based on the load size and the autoclave manufacturer’s suggested timing. Your comment made us realize that this was too detailed, and we have now changed the sentence to: “Approximately 1 g of glass wool was autoclaved in a 500 mL Erlenmeyer flask loosely covered with aluminum foil.”
Line 247-248 corrected text now reads: “After 24 hours, the supernatant in the flask was removed using serological pipettes, re-placed with fresh media and returned to the shaking incubator.”
Line 251: Replaced “swimming” with “planktonic”
Reviewer Comment: “state the device used for lyophilization and the woriking conditions”
The corrected text now reads: “Samples were frozen with liquid nitrogen and lyophilized (VirTis, -55 °C, 20 mTorr, overnight) to remove water.”
Line 282 corrected text now reads: “Samples were then stored at room temperature (20 – 22 °C).”
Line 331 (in the Reviewer’s document) was removed as suggested.
Results Section
We ensured that E. coli was italicized throughout our manuscript.
Line 336 and 337: “swimming” replaced with “planktonic”
Line 361: Table 1 has been reformatted to use the footnote function. The formatting of the table conforms to the journal’s requirements.
Line 382 -384 corrected text now reads: “Arabinose addition in the presence of levofloxacin recovered biofilm colony growth at both temperatures (Figure 2A panels 25 ng/mL Lev vs. 0.5% Ara + 25 ng/mL Lev). Colony growth with both arabinose and levofloxacin added was not significantly different from the control but smaller than the observed growth with arabinose alone (Figure 2B).”
Line 561: corrected “changes” to “differences”
Discussion Section:
Line 592 corrected text now reads: “Moreover, we discovered that the presence of arabinose enhanced survival of E. coli cells in the biofilm state when challenged with levofloxacin, a phenomenon we had previously only observed for cells in the planktonic state [16].”
Reviewer 2 Report
Comments and Suggestions for Authors
Minor comments
This study investigates the impact of L-arabinose on Escherichia coli biofilm formation and antibiotic susceptibility. L-arabinose enhanced biofilm growth and reduced levofloxacin efficacy, particularly at 37 °C. Transcriptomic analyses revealed temperature- and state-dependent gene expression changes involving efflux pumps, transporters, and biofilm-associated structures. Key metabolic and regulatory genes were modulated in response to L-arabinose, suggesting altered physiological states. These findings underscore the importance of environmental sugars in modulating antibiotic resistance in bacterial biofilms.
[Lines 16–17]: The authors report that L-arabinose reduces levofloxacin efficacy and promotes E. coli growth under sub-inhibitory antibiotic concentrations. Could the authors elaborate on the precise concentration-dependent changes in MIC and tolerance, and whether these were validated via standard CLSI protocols?
[Lines 22–23]: The study highlights transcriptomic modulation of antibiotic resistance and biofilm-associated genes. Which specific genes related to pili, curli, and efflux transporters were consistently upregulated or downregulated across experimental conditions?
[Lines 91–92]: The study employs the E. coli PHL628 strain, possessing an ompR mutation that constitutively induces biofilm formation. How does this genetic background influence the generalizability of findings to wild-type clinical isolates?
[Lines 102–106]: A 10% (w/w) L-arabinose stock solution was utilized in multiple assays. Was the stability of this stock confirmed over the experimental timeline, and were pH or osmolarity effects assessed?
[Lines 125–130]: The authors used 25 ng/mL levofloxacin in growth assays. On what basis was this concentration selected, and does it represent a sub-MIC level for PHL628 under the specified growth conditions?
[Lines 140–144]: Glass wool was selected as the substratum for biofilm cultivation. Could the authors clarify its physicochemical relevance and surface properties compared to conventional polystyrene or glass surfaces?
[Lines 153–157]: In the crystal violet assay, 30% acetic acid was employed for dye solubilization. Were any controls implemented to assess whether this extraction method affected OD590 absorbance accuracy or introduced variability?
[Lines 183–188]: The EPS extraction protocol involved salt treatment and vortexing. How was consistency in EPS recovery ensured across conditions, and was the extraction efficiency validated?
[Lines 219–221]: Both calcofluor white and SYPRO Ruby staining were applied to analyze EPS components. Could the authors provide justification for using both stains, and discuss whether differential sensitivity influenced interpretation?
[Lines 283–302]: RNA integrity is critical for transcriptomic analyses. Were RNA quality metrics such as RIN scores assessed prior to sequencing, and how many biological replicates per condition were used to ensure statistical power?
[Lines 317–325]: The threshold for identifying differentially expressed genes was set at logâ‚‚FC > 2 and adjusted p-value < 0.01. Was this cutoff selected based on biological relevance or statistical power analysis?
[Lines 475–479]: Among the consistently upregulated genes were toxin-antitoxin modules and stress response regulators. Were any of these genes functionally validated through gene deletion, complementation, or overexpression to confirm their phenotypic roles?
[Lines 540–545]: Pilus-associated gene expression varied by temperature and growth state. What criteria were used to differentiate between structural fimbrial genes (e.g., fimA, fimH) and those encoding regulatory or accessory components?
[Lines 574–584]: Several efflux transporters were differentially regulated in response to L-arabinose. Could the authors discuss whether the observed expression patterns of MFS or RND family pumps may contribute to reduced levofloxacin susceptibility?
[Lines 667–676]: The authors suggest that metabolic context, such as sugar availability, alters antibiotic response. Could these findings inform the design of therapeutic strategies targeting biofilm-associated infections in nutrient-variable environments?
Author Response
We thank the reviewer for their time and effort in reviewing our paper. We address their questions and comments below.
[Lines 16–17]: The authors report that L-arabinose reduces levofloxacin efficacy and promotes E. coli growth under sub-inhibitory antibiotic concentrations. Could the authors elaborate on the precise concentration-dependent changes in MIC and tolerance, and whether these were validated via standard CLSI protocols?
We use 25 ng/mL levofloxacin in this study, as we had found the MIC and MDK for the antibiotic with our cell line as previously reported (Ref. 16, Frizzell et al. 2024), for which we did use standard CLSI protocols. To add information to the abstract, this line in the abstract has been updated to: “L-arabinose reduced the efficacy of levofloxacin and promoted growth in sub-minimum inhibitory concentrations (25 ng/mL).”
[Lines 22–23]: The study highlights transcriptomic modulation of antibiotic resistance and biofilm-associated genes. Which specific genes related to pili, curli, and efflux transporters were consistently upregulated or downregulated across experimental conditions?
We added information about specific genes to this sentence in our abstract, which now reads: “Using gene ontology overrepresentation analyses, we found that L-arabinose modulated the expression of many critical antibiotic resistance genes, including efflux pumps (ydeA, mdtH, mdtM), transporters (proVWX), and biofilm-related genes for external structures like pili (fimA) and curli (csgA, csgB).”
[Lines 91–92]: The study employs the E. coli PHL628 strain, possessing an ompR mutation that constitutively induces biofilm formation. How does this genetic background influence the generalizability of findings to wild-type clinical isolates?
This is an excellent question. We are uncertain of the generalizability of our results to other, clinical isolates, but we are interested in pursuing this in the future. Our laboratory has already begun investigating the probiotic Nissle 1917 strain and, anecdotally, we have noticed some differences in our biofilm assay results. For the scope of this paper and the amount of data we are presenting, we were interested in first focusing on a single strain.
[Lines 102–106]: A 10% (w/w) L-arabinose stock solution was utilized in multiple assays. Was the stability of this stock confirmed over the experimental timeline, and were pH or osmolarity effects assessed?
Stock solutions are prepared fresh daily/for each experiment. We did not assess the stability in our media during the longest timepoints (e.g., 48 h for biofilm growth on agar), though we note that Sigma Aldrich (our supplier), states that L-arabinose has a stability at room temperature for 6 months (https://www.sigmaaldrich.com/US/en/product/mm/178680). We have a separate manuscript in preparation that examines the effect of arabinose and other sugars on the external pH (as measured by a pH meter) and internal pH (as measured by a pH-sensitive GFP) of our cultures—a project that was stimulated by our RNA-seq results that showed increased expression of acid stress resistance genes. We have not measured how the osmolarity of our solutions change with the addition of arabinose, but, based on the LB we use (Lennox, 235-261 mOsm/L) our 0.5% (w/w) arabinose solutions (33 mM, 33mOsm/L) would be changing the osmolarity by about 15%.
[Lines 125–130]: The authors used 25 ng/mL levofloxacin in growth assays. On what basis was this concentration selected, and does it represent a sub-MIC level for PHL628 under the specified growth conditions?
We use 25 ng/mL levofloxacin in this study, as we had found the MIC to be 50 ng/mL for the antibiotic with our cell line as previously reported (Ref. 16, Frizzell et al. 2024).
[Lines 140–144]: Glass wool was selected as the substratum for biofilm cultivation. Could the authors clarify its physicochemical relevance and surface properties compared to conventional polystyrene or glass surfaces?
Glass wool provides a hydrophilic surface like a glass slide with considerable surface area and sites of nucleation where biofilms grow. We have published this protocol previously (Ref 12. Hantus et al. 2024) and are using it identically. We adopted this protocol from this paper (now Ref 18): (Hayette Benamara, Christophe Rihouey, Thierry Jouenne, Stéphane Alexandre, Impact of the biofilm mode of growth on the inner membrane phospholipid composition and lipid domains in Pseudomonas aeruginosa. Biochimica et Biophysica Acta (BBA) - Biomembranes,
Volume 1808, Issue 1, 2011, Pages 98-105, ISSN 0005-2736, https://doi.org/10.1016/j.bbamem.2010.09.004.
(https://www.sciencedirect.com/science/article/pii/S0005273610003196).
We now also cite this paper here on Line 136-137: “Biofilm was grown with 0.2 g of sterile glass wool (Sigma-Aldrich) in a clear 6-well plate (VWR, Radnor, PA, USA) as described previously [12,18,19].”
[Lines 153–157]: In the crystal violet assay, 30% acetic acid was employed for dye solubilization. Were any controls implemented to assess whether this extraction method affected OD590 absorbance accuracy or introduced variability?
Early glass wool experiments in our lab employed negative controls (no cells added, only LB), where we saw less than 6% of the absolute absorbance value of the 0% sugar condition (e.g., average OD595 values for 0% were around 0.5, we observed values of 0.03, which is also similar to the absorbance of our acetic acid blank.)
[Lines 183–188]: The EPS extraction protocol involved salt treatment and vortexing. How was consistency in EPS recovery ensured across conditions, and was the extraction efficiency validated?
We do not have an orthogonal mechanism (e.g., LC/MS-MS) for confirming extraction of the EPS other than our spectroscopic biochemical assays and confocal microscopy. We employed at least three biological replicates per condition and using this robust extraction technique led to similar biomolecule concentrations for a given condition.
[Lines 219–221]: Both calcofluor white and SYPRO Ruby staining were applied to analyze EPS components. Could the authors provide justification for using both stains, and discuss whether differential sensitivity influenced interpretation?
These two stains examine two different biomolecules. SYPRO Ruby binds to extracellular proteins and reflects the protein content within the EPS. Calcofluor white binds to polysaccharides and reflects the carbohydrate content of the EPS. As they are reporting on two different types of molecules, we do not compare the stains to one another, as they are inherently different.
[Lines 283–302]: RNA integrity is critical for transcriptomic analyses. Were RNA quality metrics such as RIN scores assessed prior to sequencing, and how many biological replicates per condition were used to ensure statistical power?
We thank the reviewer for making this point and agree. The company that performed our RNA-seq data collection, Mr.DNA, did run RNA integrity testing and performed analyses only on samples that had an RIN of 7 or higher. Three biological replicates were prepared for each condition. We have updated the text to reiterate these important points:
Lines 279-280: “Three biological replicates were prepared for each experimental condition.”
Lines 286-288: “The concentration of the RNA was determined using the Qubit® RNA Assay Kit (Life Technologies) and the RNA integrity number (RIN) was determined; all samples had a RIN greater than 7.”
[Lines 317–325]: The threshold for identifying differentially expressed genes was set at logâ‚‚FC > 2 and adjusted p-value < 0.01. Was this cutoff selected based on biological relevance or statistical power analysis?
We thank the reviewer for their question. This cutoff was selected based on biological relevance, as we were interested in large/dramatic changes in gene expression.
[Lines 475–479]: Among the consistently upregulated genes were toxin-antitoxin modules and stress response regulators. Were any of these genes functionally validated through gene deletion, complementation, or overexpression to confirm their phenotypic roles?
We thank the reviewer for their question. These are excellent next experiments which we aim to conduct in our laboratory but are outside of the scope of this already data-dense manuscript. We are particularly interested in the deletion and overexpression of the HipA/HipB toxin-antitoxin system as it is downregulated in biofilm but upregulated in planktonic cells at 37 ºC with arabinose.
[Lines 540–545]: Pilus-associated gene expression varied by temperature and growth state. What criteria were used to differentiate between structural fimbrial genes (e.g., fimA, fimH) and those encoding regulatory or accessory components?
In our original analysis we did not differentiate between the structural vs. regulatory components of the fim operon. Inspired by the reviewer’s question, we note that the structural fimbrial gene fimA and the putative fimbrial gene fimI more exaggerated in their upregulation (at 28 C) and downregulation (at 37 C) than the other fimbrial proteins. The genes fimF and fimG both code for adaptor proteins that connect the primary fimbrial rod to the FimH adhesin protein at the tip. We have added some of these details to our discussion in lines 644 – 647: “We observe the upregulation of the structural type I fimbriae gene fimA and the fimF and fimG genes, the products of which act as adaptors between the fimbrial rod and the FimH tip at lower growth temperatures, which are overexpressed in uropathogenic E. coli isolates that demonstrate resistance to multiple classes of antibiotics [59].”
[Lines 574–584]: Several efflux transporters were differentially regulated in response to L-arabinose. Could the authors discuss whether the observed expression patterns of MFS or RND family pumps may contribute to reduced levofloxacin susceptibility?
We thank the reviewer for this question and have added some additional detail and clarifying language to make the connection between the increased expression of these pumps and reduced levofloxacin susceptibility. Line 660-662: “We also observed upregulation of resistance-nodulation-division (RND) pump genes mdtE and mdtF. Any of these genes could be contributing to reduced sensitivity of E. coli to levofloxacin while in the presence of arabinose by increasing levofloxacin efflux.”
[Lines 667–676]: The authors suggest that metabolic context, such as sugar availability, alters antibiotic response. Could these findings inform the design of therapeutic strategies targeting biofilm-associated infections in nutrient-variable environments?
We thank the reviewer for their question. We certainly think that understanding how simple nutrients like sugars impact antibiotic sensitivity and biofilm formation could lead to therapeutic handles for further development. We end our manuscript with this newly updated sentence Line 677-680: “These results demonstrate a previously unrecognized role for L-arabinose in shaping antibiotic response in E. coli and suggest that metabolic context should be carefully considered in efforts to combat biofilm-associated infections and in the development of the next generations of therapeutics.”
Round 2
Reviewer 1 Report
Comments and Suggestions for Authors
Thank you for the revised version